# Electrochromic windows with fast response and wide dynamic range for visible-light modulation without traditional electrodes

Zhuofei Jia[1,3], Yiming Sui[1,3], Long Qian[1,3], Xi Ren[1], Yunxiang Zhao[1], Rui Yao[1], Lumeng Wang[1], Dongliang Chao [2] & Cheng Yang [1] ✉

Electrochromic (EC) devices represent an emerging energy-saving technology, exhibiting the capability to dynamically modulate light and heat transmittance. Despite their promising potential, the commercialization of EC devices faces substantial impediments such as high cost, intricate fabrication process, and low optical contrast inherent in conventional EC materials relying on the ion insertion/extraction mechanism. In this study, we introduce an innovative "electrode-free" electrochromic (EC) device, termed the EECD, which lacks an EC-layer on the electrodes during device assembling and in the bleached state. This device features a simplified fabrication process and delivers superior optical modulation. It achieves a high optical contrast ranging from 68-85% across the visible spectrum and boasts a rapid response time, reaching 90% coloring in just 17 seconds. In addition, EECD exhibits stable cycling for over 10,000 cycles without noticeable degradation and maintains functionality across a broad temperature range (0 °C to 50 °C). Furthermore, the fabricated large-area devices (40 cm × 40 cm) demonstrate excellent tinting uniformity, suggesting excellent scalability of this approach. Our study establishes a paradigmatic breakthrough for EC smart windows.

Electrochromic (EC) windows, heralded for their role in enhancing building energy efficiency and elevating residential living, stand at the forefront of groundbreaking technology[1,2]. To count, there are approximately 260 billion m² of building stock globally with a 2% annual growth rate, while the average retrofit rate of decarbonization of building stock is currently only 1% per year[3,4]. Undoubtedly, introducing EC windows to replace conventional window components such as shutters or curtains in the existing buildings is conducive to accelerating the realization of zero-carbon-ready-buildings, i.e., highly energy-efficient and resilient buildings[5,6], yet the high price greatly reduces people's motivations to do so[7,8]. Particularly, when analyzing a classic sandwich-structured EC devices, transparent conductive oxide (TCO) glass and electrolyte are generic materials with negligible costs,

whereas the EC-layer, serving as the core function of the device, constitutes the primary cost, making the production costs of EC windows persistently remain above 180–250 USD m⁻²[9]. Regrettably, even though the existing EC materials exhibit less-than-ideal performance, their production still involves sophisticated processes e.g., multilayer magnetron sputtering; thus, setting a high threshold for further cost reduction. In light of this, reducing the manufacturing cost of EC windows down to the average consumer's cost expectations for curtains (< 60 USD m⁻², in line with 90% of commercial curtain prices on Amazon) is a key node toward their mass deployment, while simultaneously ensuring outstanding EC performance and easy installation. For the prevailing EC materials, such as metal oxides (e.g., $WO_3$ and $NiO_x$) and organic molecules, which change color through charge

[1]Institute of Materials Research, Tsinghua Shenzhen International Graduate School, Tsinghua University, Shenzhen 518055, P.R. China. [2]Laboratory of Advanced Materials, Shanghai Key Laboratory of Molecular Catalysis and Innovative Materials, State Key Laboratory of Molecular Engineering of Polymers, College of Chemistry and Materials, Fudan University, Shanghai 200433, P. R. China. [3]These authors contributed equally: Zhuofei Jia, Yiming Sui, Long Qian. ✉e-mail: yang.cheng@sz.tsinghua.edu.cn

insertion/extraction, the challenges lie in conductivity and light modulation range[10,11]. So, there are stringent requirements for the EC coating concerning density, thickness, uniformity, and crystal structure. Inevitably, the prevalent approach involves thick, multi-layer magnetron sputtering coatings to achieve a high-quality EC coating, making it difficult to further reduce the marginal cost for mass production[12].

Recently, metallic materials, e.g., Bi[13], Cu[14], and Zn[15], have emerged as a new group of EC materials featuring the electrodeposition-dissolution mechanism. The metallic electrodes enable spectral transmittance tuning across a broader spectral range, from <0.01% at the plating state to 80–90% transmittance at the stripping state[12,16,17]. However, single-EC-layer devices based on metal plating/stripping usually suffer from a long response time, up to several minutes to reach fully opaque states, and short cycle life caused by non-durable

metal mesh counter electrodes[18,19]. Several strategies have been proposed to address these issues, including employing hybrid windows that combine metal-based working electrode with the ion-insertion/extraction EC counter electrode or decorating substrate surfaces with noble metal to facilitate metal nucleation[20], but this undoubtedly will again raise the complexity of preparation, leading to high cost and limit the EC windows availability. Essentially, there is still a lack of scalable and cost-effective methods to propel their widespread application and achieve significant socio-economic value (cost reduction and process simplification).

Herein, we report an "electrode-free" EC device (EECD) strategy, featuring two pristine TCO glasses and a simple injection of aqueous electrolyte, which showcase a minimalist design (Fig. 1a) and state-of-the-art EC performance. This approach adds convenience and flexibility, requiring minimal processing steps after cutting the TCO and

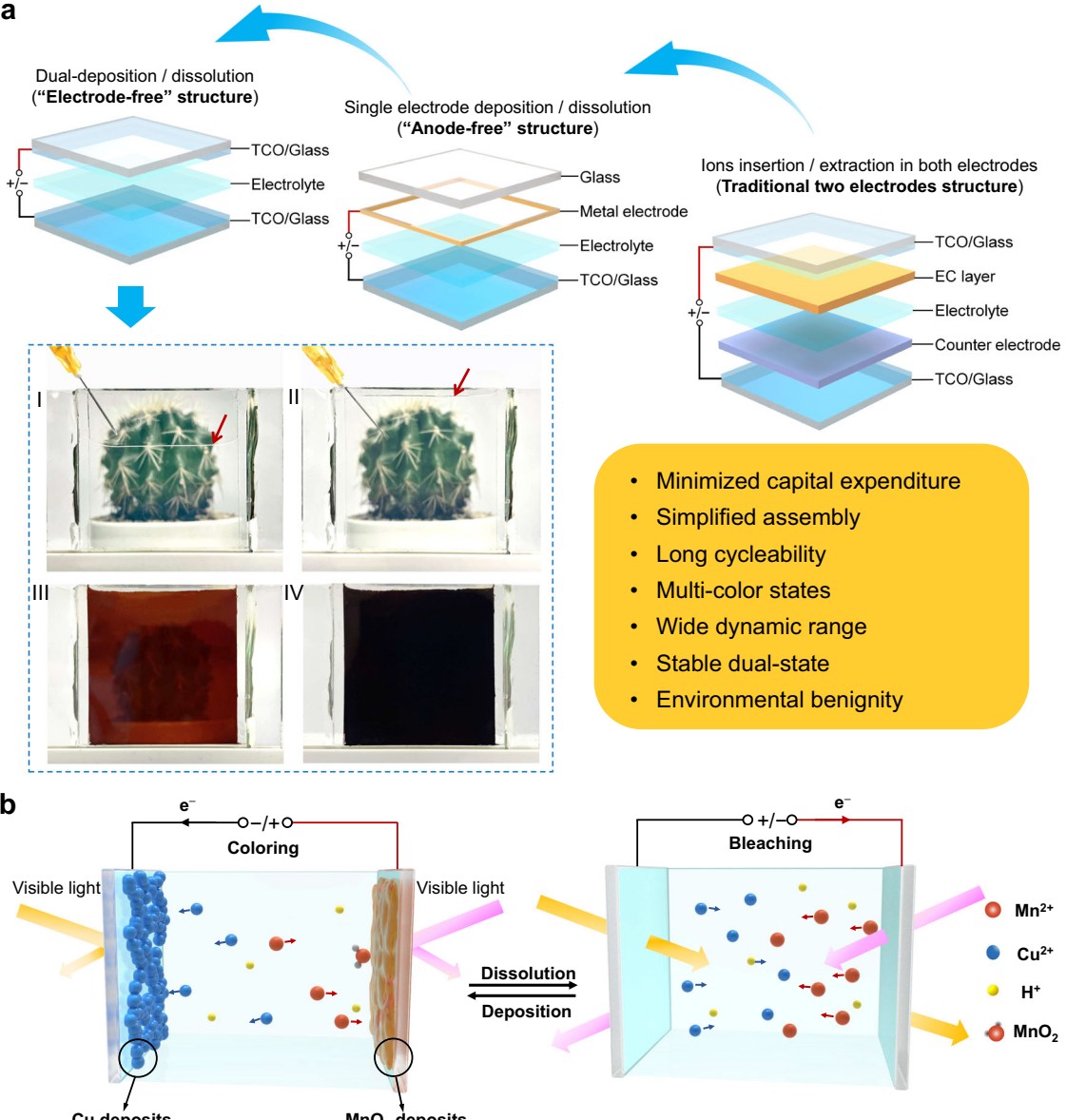

**Fig. 1 | Schematic structure and working mechanism of "electrode-free" electrochromic device (EECD). a** The structures of three types of electrochromic (EC) devices in the sequence of increased fabrication complexity, from conventional EC device containing both EC-layer and counter electrode (right) to "anode-free" EC device with single counter electrode (middle), and to EECD with no electrodes (left). The photographic images of EECD were taken during the electrolyte injection process (I and II, where the red arrows point toward the liquid levels) and at different colored states after applying 1.6 V (III and IV). **b** The working mechanism of EECD featuring the MnO$_2$/Mn$^{2+}$ electrochromic redox at the cathode and the Cu$^{2+}$/Cu electrochromic redox at the anode. The fluorine-doped tin oxide (FTO)/glasses were employed as both electrodes and a colorless electrolyte is sandwiched between the two electrodes.

even users can easily inject the electrolyte between two pieces of TCO glass to enable EC function by "do it yourself". The working mechanism of EECD relied on dual deposition/dissolution on both cathode and anode. Notably, EECD synergistically achieves high optical contrast, fast response, long stability, and large-area. It means that the exceptional EC performance can be achieved through electrolyte engineering without intricate processing. The EECD paradigm utilizes $MnO_2/Mn^{2+}$ for the cathode and $Cu^{2+}/Cu$ for the anode. The complementary absorption spectra of both electrodes enable EECD to tune transmittance across a broad visible spectrum, achieving < 0.01% transmittance and fast response (coloring 90% in 17 s). The low conductivity of deposited $MnO_2$ ensures homogenizing the ion flux and electric field in the EECD, which embodies a unique "self-correction" ability for color uniformity. Additionally, it delivers excellent stability for over 10,000 cycles. Last but not least, the "electrode-free" structure enables the easy fabrication of large-area devices (40 cm × 40 cm), indicating the potential for scalable production.

## Results

### Electrochromism performance of electrolytic metal oxides

In elucidating the working mechanism of EECD, as illustrated in Fig. 1b, the cathode undergoes the reversible cycling between solvated metal ions and solid metal oxide. In this regard, we selected the $MnO_2/Mn^{2+}$ redox pair, where $MnO_2$ exhibits diverse advantages, including cost effectiveness, environmental friendliness, and inherent physicochemical stability. In addition, the $MnSO_4$ aqueous solution is colorless and $MnO_2$ displays a dark brown color, fulfilling the demands of electrochromism (Supplementary Fig. 1).

Before fabricating a full device, we assessed the EC performance of the $MnO_2/Mn^{2+}$ reaction by analyzing the ultraviolet-visible (UV-Vis) transmittance spectra at 460 nm of a series of electrolytes containing 0.5 M $MnSO_4 + x$M $H_2SO_4$ (x = 0, 0.3, 0.5, 1). In this acidic condition, we select fluorine-doped tin oxide (FTO) glass as a transparent conductive substrate due to its excellent stability in acidic solutions[19,21]. Notably, the addition of $H_2SO_4$ over 0.5 M increases the transmittance recovery from 30% to 92%. In addition, we found that increasing the $H_2SO_4$ concentration from 0.3 M to 1.0 M shortened the dissolution time from 721 s to 224 s (Supplementary Fig. 2). These findings suggest that the introduction of $H_2SO_4$ can facilitate the dissolution of $MnO_2$. But the $MnO_2$ deposition rate becomes more sluggish in acidic environments, as evidenced by the longer coloring time and lower deposition currents. To strike a balance, we chose 0.5 M $H_2SO_4$ + 0.5 M $MnSO_4$ as the optimal composition for further investigation (Supplementary Figs. 3, 4).

The window color is a crucial factor to consider. Notably, the deposition and dissolution of $MnO_2$ displayed distinctive color variations ranging from light yellow to dark brown and to black when the $MnO_2$ thickness increased from ca. 200 nm to 1000 nm (Fig. 2a). It is worth noting that the thicknesses of deposited $MnO_2$ were determined through the cross-sectional scanning electron microscopy (SEM) (Supplementary Fig. 5). Furthermore, the energy-dispersive X-ray spectrum (EDX) results demonstrated the uniform deposition of $MnO_2$ on the FTO electrodes (Supplementary Figs. 6, 7). In order to render the 'perceived' color for each optical state, we employed a spectrophotometer to qualitatively analyze the colors using three parameters (CIE 1931): $L^*$, $a^*$, and $b^*$, where $L^*$ denotes lightness while $a^*$ and $b^*$ represent the ratios of green-red and blue-yellow colors, respectively (Fig. 2b and Supplementary Table 1). At the fully bleached state, the ($L^*$, $a^*$, $b^*$) coordinates are (93.74, 0.04, 2.25), and they shift to (4.81, 1.34, −0.02) at the fully colored state, indicating the transition between colorless and black. To accurately identify the color change, we transformed the ($L^*$, $a^*$, $b^*$) coordinate to the $x^*y^*$ color space (CIE 1931), where the x-axis represents the proportion of red primary color, the y-coordinate denotes the ratio of green primary color, and the

z-coordinate represents the ratio of blue primary color, which derives from the equation: $x + y + z = 1$.

As shown in Fig. 2c, the electrode displays a brown-yellow color when the deposition proceeds within 180 s, resulting in a selective modulation towards blue and violet light from 400 to 600 nm. Furthermore, the color of $MnO_2$ approaches the black color at higher mass loadings, thereby expanding the adsorption lights to higher wavelengths. Therefore, the electrode transmittance continuously decreases as deposition time extends (Fig. 2d)[22]. Notably, the electrode achieves a low transmittance of 0–5% across the visible light range (400–800 nm) at 600 s, enabling a wide transmittance modulation range of the $MnO_2$ EC electrode, namely a substantial optical contrast (ΔT). Figure 2e illustrates the optical response curve of $MnO_2$ deposition/dissolution on FTO electrodes, demonstrating a high ΔT of ca. 93% between the fully colored and bleached states. It is worth noting that we selected the wavelength of 460 nm to assess the EC performance of $MnO_2/Mn^{2+}$. This selection is based on two main factors: the most pronounced change in transmittance observed at this wavelength and the safety concerns associated with the blue light spectrum[23,24]. For practical considerations, we defined the time consumed to reach 90% complete coloring and bleaching states as the coloring time ($t_c$) and the bleaching time ($t_b$), which are 37.7 s and 153.2 s, respectively. To assess the stability of the $MnO_2/Mn^{2+}$ reaction on FTO electrodes, we charged it at 1.6 V for 30 s and then discharged it at 0.2 V for 120 s. The electrode maintains a high optical contrast at 460 nm and fast response time after 2000 cycles without degradation (Supplementary Fig. 8–10).

To elucidate the mechanism of $MnO_2$ deposition and dissolution, we performed both the cyclic voltammetry (CV) test and in situ UV-Vis transmittance measurement. As shown in Fig. 3a, a single sharp anodic peak at 1.4 V corresponds to the oxidation of $Mn^{2+}$ to $MnO_2$, resulting in a rapid decrease in transmittance. X-ray diffraction (XRD) result confirmed the predominant γ-$MnO_2$ in the deposited material (JCPDS #14-0644) (Fig. 3e, black line). In contrast, two major cathodic peaks at 1.2 V and 0.4 V, along with a minor peak at 1.0 V, revealed a complicated mechanism in $MnO_2$ dissolution. Interestingly, the transmittance slightly changes in stage I (1.3–1.1 V) and II (1.1–0.7 V) but dramatically increases in stage III (0.7–0.2 V). Furthermore, we observed three plateaus in the galvanostatic discharging test, consistent with the CV results. The result of the electrodes prepared by galvanostatic discharging to 1.1 V in stage I, 0.7 V in stage II, and 0.4 V in stage III was analyzed (Supplementary Fig. 11). To probe the variance of chemical states, we performed the ex situ X-ray photoelectron spectroscopy (XPS) measurements for both Mn and O elements (Fig. 3b, c). The XPS O 1s spectrum of the pristine $MnO_2$ electrode is deconvoluted to three peaks, i.e., Mn–O–Mn at -530.2 eV, Mn–O–H at -531.5 eV, and H–O–H at -533.3 eV[25]. In stage I, no significant change was observed in other bond peaks except for a weaker signal of Mn–O–Mn, conforming with the reduced peak intensity of $Mn^{4+}$ ions in the XPS Mn 2p spectrum. These results indicate the partial dissolution of $MnO_2$ into the electrolyte in stage I via Eq. (1)[26,27]:

$$MnO_2 + 4H^+ + 2e^- \rightarrow Mn^{2+} + 2H_2O \qquad (1)$$

To gain insight into the electrolytic reactions, the cathodic processes of $MnO_2$ electrodes with different mass loadings were analyzed. Noteworthily, we found that the current responses in stage I were close in the LSV profiles, in sharp contrast to the progressively stronger peaks in stage III (Supplementary Fig. 12). Therefore, we postulated that the reduction reaction in stage I primarily happened on the surface of $MnO_2$, where it contains more incomplete-coordinated reactive oxygen species, i.e., abundant Mn vacancies[27–29]. The instability of the surface state is conducive to the reduction of near-surface $Mn^{4+}$,

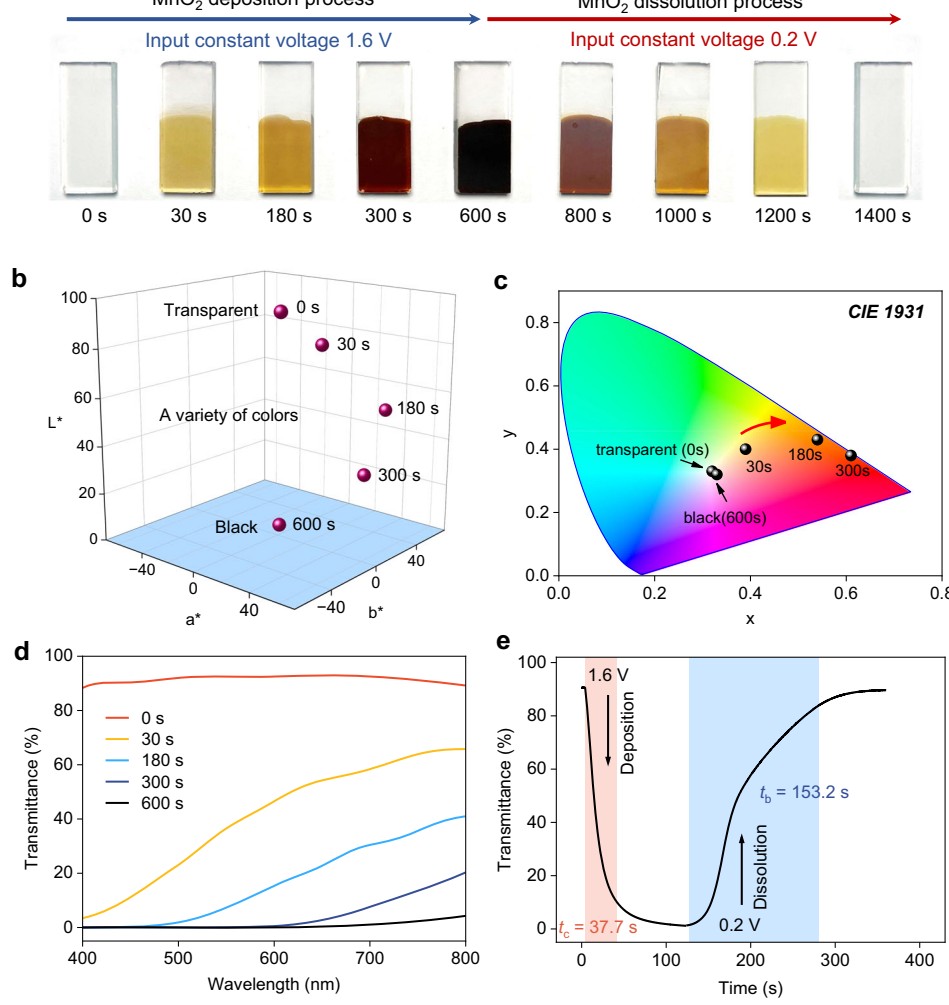

**Fig. 2 | Electrochromic performance of the MnO₂/Mn²⁺ redox reaction.**
**a** Photographs of MnO₂/FTO electrode, featuring color changes at different deposition/dissolution stages. **b** CIE L*a*b* color coordinates of MnO₂/FTO electrode at various charging time. **c** Peak x, y color space (CIE 1931) results of MnO₂/FTO electrode at different charging times (0 s to 600 s). **d** UV-Vis transmittance spectra of MnO₂/FTO electrode charging at a potential of 1.6 V vs. standard hydrogen electrode (SHE) for 0 s, 30 s, 180 s, 300 s, and 600 s. **e** A plot illustrating the transmittance at 460 nm over time, with the application of deposition and dissolution potentials at 1.6 V and 0.2 V, respectively.

resulting in a higher reduction potential than that of bulk Mn⁴⁺. In both stage II and stage III, we observed more intense Mn–O–H peaks and higher ratios of Mn³⁺ in the O 1s and Mn 2p spectra, suggesting the successive insertion of H⁺ into the oxides lattice (Fig. 3b, c). The electron energy-loss spectroscopy (EELS) revealed that the Mn–L-edge shifts to a lower oxidation energy loss at lower discharging potentials (Fig. 3d). In addition, the energy gap between L₂ and L₃ edges (ΔE) increased from 11.7 to 12.9 and the intensity ratio of Mn–L₂,₃ edge decreased from 1.35 to 2.7. These results demonstrated a constantly decreased valence state of the product upon the proton insertion. Furthermore, we conducted XRD and high-resolution transmission electron microscopy (HRTEM) to investigate the phase evolution (Fig. 3e, f). In stage II and III, a new product appeared with the XRD pattern fitted to HMnO₂ (JCPDS 18-0804) while the lattice fringes of 0.46 nm corresponding to the d spaces of the (002) plane in tetragonal HMnO₂. The unpronounced variations in XRD peaks relate to the slight distortion of the interplanar spacing of MnO₂ during H⁺ insertion. The results demonstrated the intercalation of proton from the electrolyte in both stage II and III, turning Mn(IV)O₂ into HMn(III)O₂. Notably, the multi-step proton insertion may also relate to the complex structure of γ-MnO₂[30,31], in which the tunnels with sizes of 2.3 Å × 4.6 Å [1 × 2] and 2.3 Å × 2.3 Å [1 × 1] are randomly distributed (Supplementary Fig. 13)[32].

Lastly, the HMnO₂ is constantly consumed in stage III during proton insertion via Eq. (2)[33].

$$HMnO_2 + 3H^+ + e^- \rightarrow Mn^{2+} + 2H_2O \qquad (2)$$

which is confirmed by the significantly increased transmittance (Fig. 3a). (The schematic of dissolution mechanism of the MnO₂ electrode is illustrated in Supplementary Fig. 14.)

Furthermore, we found that some other metal oxides also exhibit reversible EC performance based on the deposition/dissolution chemistry, e.g., PbO₂. The PbO₂/Pb²⁺ chemistry displayed a uniform tinting and bleaching process on FTO glass and the PbO₂/FTO electrode showed a similar color change to that of MnO₂ (Supplementary Fig. 15). On the other hand, the PbO₂/Pb²⁺ redox pair demonstrated some different properties from the MnO2/Mn²⁺ pair. For example, the deposition and dissolution of PbO₂ occurs at a lower potential range between −0.4 V and 0.4 V than the MnO₂ counterpart. More importantly, different from the MnO₂/Mn²⁺ redox pair, the PbO₂/Pb²⁺ redox pair only displays one cathodic peak in the CV curve, referring to the direct reduction of PbO₂ into Pb²⁺. Such characteristics have a close relationship with the fast and highly reversible tinting and bleaching processes of the PbO₂/Pb²⁺ redox (Supplementary Fig. 16). Therefore,

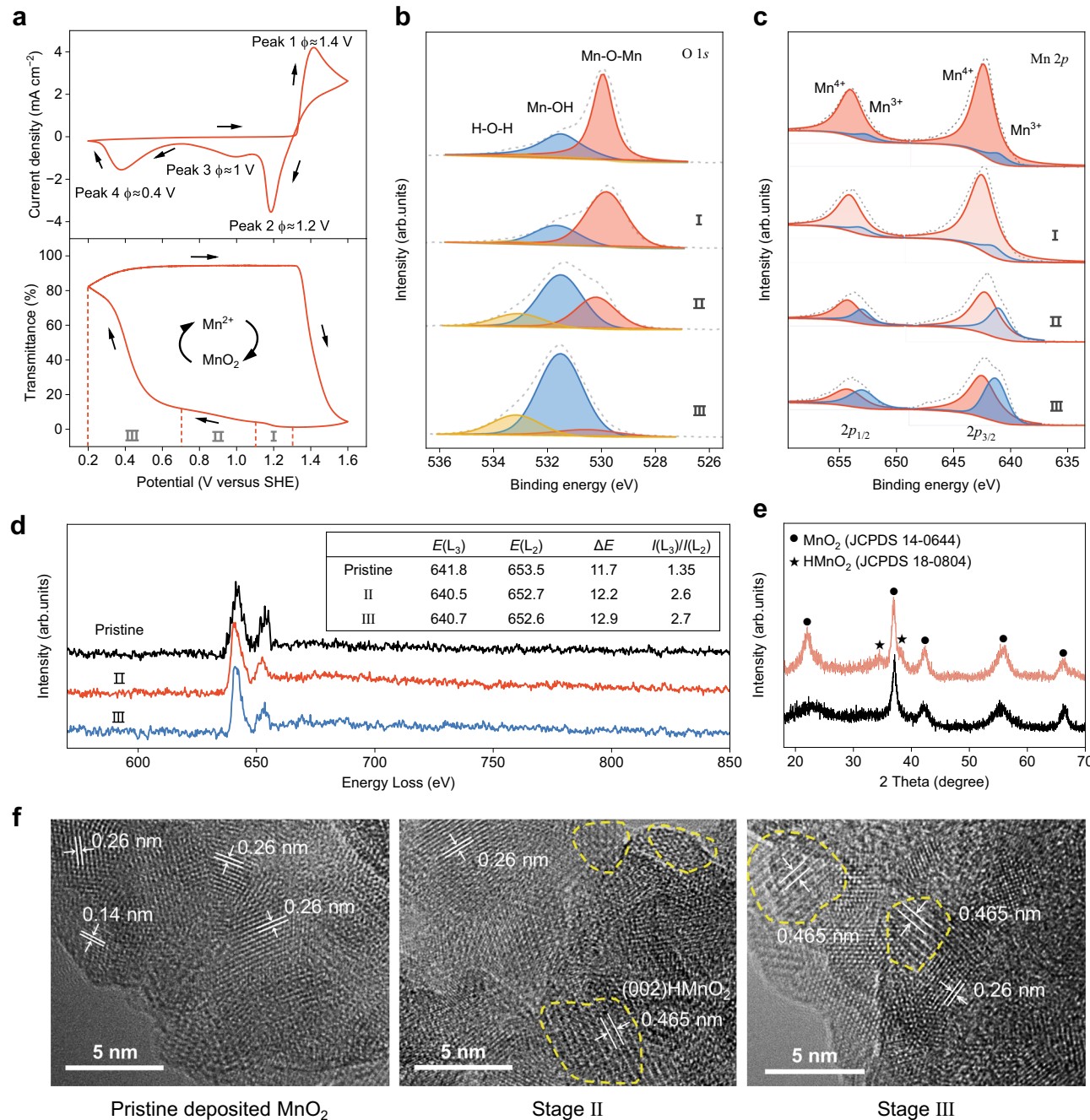

**Fig. 3 | Deposition/dissolution chemistry of the MnO₂/Mn²⁺ redox on FTO.**
**a** Cyclic voltammetry (CV) curve and in situ UV-Vis transmittance spectrum at 460 nm of the electrolytic MnO₂/Mn²⁺ reaction in a 0.5 M MnSO₄ + 0.5 M H₂SO₄ electrolyte at a scan rate of 5 mV s⁻¹. **b** XPS spectra of O 1s from deposited MnO₂ electrodes to the different discharging potentials of I (1.1 V), II (0.7 V), and III (0.4 V). **c** XPS spectra of Mn 2p from deposited MnO₂ electrodes at different discharging potentials of I (1.1 V), II (0.7 V), and III (0.4 V) **d** EELS spectra analyzing Mn valence for the pristine and discharged to II (0.7 V) and III (0.4 V) samples, respectively. **e** XRD patterns of the MnO₂ electrode at the pristine state (black) and being discharged to 0.7 V (red). **f** HRTEM images of the as-deposited MnO₂ electrode sample, and the ones being discharged to II (0.7 V) and III (0.4 V).

more oxides featuring the deposition/dissolution chemistry should be explored as EC materials, where their distinctive behaviors provide opportunities for developing more advanced EECDs.

## Electrochromism performance of the Cu²⁺/Cu plating/stripping pair

As for the anode, the Cu²⁺/Cu redox pair was chosen due to the excellent anti-corrosion stability of Cu metal in acidic electrolytes and the high optical extinction coefficient of copper[34]. To maintain high light transparency, only 0.1 M CuSO₄ was added into the 0.5 M

MnSO₄ + 0.5 M H₂SO₄ electrolyte. CV diagram and in situ UV-Vis transmittance spectra were analyzed on the pristine FTO electrode in a three-electrode configuration to evaluate its EC performance (Supplementary Fig. 17). It showed notable plating/stripping peaks in CV curves with a low overpotential of 0.3 V and excellent optical reversibility. XRD result demonstrated the pure phase of copper metal deposited on FTO glass (Supplementary Fig. 18). In addition, the Cu deposited on FTO electrode displayed a smooth morphology, thus facilitating the uniform coloring/bleaching of the electrode (Supplementary Fig. 19). More importantly, the Cu/FTO electrode exhibited

indiscriminate light blocking across the whole testing spectrum from 400 to 800 nm, achieving a low transmittance of ca. 0% after 180 s deposition (Supplementary Fig. 20). Furthermore, the transmittance spectra of the Cu electrode in one cycle demonstrated the $Cu^{2+}/Cu$ redox was suitable as anode chemistry in EECDs with the advantages of fast response, good reversibility, and high optical contrast (Supplementary Fig. 21).

## EECDs based on dual deposition/dissolution chemistry

The concurrent electrodeposition of $MnO_2$ on the cathode and Cu on the anode is responsible for the coloring of EECD; while in the bleaching process, the $MnO_2$ and Cu dissolve back into the electrolyte. Linear sweep voltammetry (LSV) results demonstrated that the deposition of $MnO_2$ and Cu occurs before the oxygen evolution reaction and hydrogen evolution reaction, promising a long cycling life and high reversibility of the EECD (Supplementary Fig. 22). To assess the electrochemical performance of the EECD, we tested CV with UV-Vis transmittance at a scan rate of 5 mV s$^{-1}$ in a 0.5 M $MnSO_4$ + 0.1 M $CuSO_4$ + 0.5 M $H_2SO_4$ aqueous electrolyte between −0.2 V and 1.6 V vs. $Cu^{2+}/Cu$. The CV profile resembles that of the $MnO_2$ half-cell, exhibiting one anodic peak and three cathodic peaks (Supplementary Fig. 23). In addition, the coloring occurs after the anodic peak, resulting in a rapid decline of transmittance to ca. 0%, while the bleaching gradually commences after the first cathodic peak at 1.0 V and finally recovers to a transparent state after all three peaks.

As shown in Fig. 4a, the EECD outperforms conventional EC devices in terms of transmittance at the bleached state of about 68-85% across the visible spectrum (400–800 nm), which is comparable to the standard low-emissivity glass window[35]. The performance originates from the EC-layer-free design and haze-reduction effect of liquid electrolyte. At the colored state, the EECD exhibited an impressively low transmittance of <0.01% across the visible spectrum (Fig. 4a, blue line). Furthermore, the EECD demonstrated a rapid and reversible optical response, with a short $t_c$ of 17 s and a corresponding $t_b$ of 147 s since the collective deposition of $MnO_2$ and Cu electrodes in EECD enhances the coloring rate (Fig. 4b). Figure 4c further illustrated that EECD delivered a shorter coloring time of 60 s than that of ca. 120 s for a single $MnO_2$ or Cu electrode to reach the 0.01% transmittance. The coloration efficiency (CE) of the EECD is about 15.3 cm$^2$ C$^{-1}$ at 460 nm (Supplementary Fig. 24). In addition, we fabricated the EC device based on $MnO_2/Mn^{2+}$ redox as the EC electrode, copper frame on transparent glass as the counter electrode, and the same electrolyte as a control EC device. The color change in this device is achieved solely through the deposition/dissolution of $MnO_2/Mn^{2+}$, and the $Cu^{2+}/Cu$ reaction does not contribute to the color change (Supplementary Fig. 25). As can be expected, this device delivered a stable EC performance.

The thermal management performance of a 100 cm$^2$ square-shape EECD was further assessed by customizing a device, wherein a black rubber plate was applied as the blackbody absorber, a xenon lamp as the light source, and an IR camera was utilized to record the temperature of blackbody in real-time (Fig. 4d, e and Supplementary Fig. 26). After exposure to an illumination equivalent to 1-Sun (100 mW cm$^{-2}$) for 90 s, a significantly lower temperature of 35 °C on the backside of the colored EECD was observed compared to that of over 50 °C on the backside of the bleached EECD. This distinct temperature contrast evidenced the superior capability of the colored EECD on thermal management.

In the idling state, the transmittance stability of EC devices in both the bleached and colored stages, namely, bistability, is crucial to their energy efficiency. The EECD samples maintained high transmittance for over 30 days in the bleached state (Fig. 4f). More importantly, the deposited $MnO_2$ and Cu can endure the acidic electrolyte for 2 h without transmittance degradation in the colored state (Fig. 4g and Supplementary Fig. 27). Therefore, the operation of EECD

is more energy-efficient compared to many reported EC windows that require constant potential to maintain the colored state[36]. Cycling durability is also a key performance parameter for evaluating EC windows. After 10,000 coloring-bleaching cycles, EECD exhibits a stable EC performance with no degradation of optical contrast and only a slight decrease in response time (Fig. 4h and Supplementary Fig. 28). We postulated that the longer response time relates to the slightly increased pH value of the electrolyte after 10,000 cycles, which originates from the inevitable hydrogen evolution in the highly acidic electrolyte (Supplementary Fig. 29). The previous analysis has shown that the concentration of protons is very important for the dissolution kinetics. When there are fewer protons, it takes longer time for bleaching. In addition, EECD can exhibit proper functioning in harsh environments, i.e., 0 °C and 50 °C, which can largely expand the application scenarios, where environmental temperature requirements are not stringent (Fig. 4i). By employing differential scanning calorimetry (DSC), the electrolyte exhibited a low freezing point of −16 °C (Supplementary Fig. 30). The decreased freezing point relative to that of pure water at 0 °C is attributed to the disrupted hydrogen bonding networks of water by introducing salts and acids[37]. Notably, the coloring performance barely changed at the various temperatures while the bleaching time increased from 145 s to 760 s when the temperature decreased from room temperature to 0 °C. This discrepancy is dictated by the different reaction kinetics of the coloring and bleaching processes. The electrochemical impedance spectroscopy (EIS) results indicate that the dissolution kinetics of $MnO_2$ are slower than its deposition, and the low-temperature conditions aggravate the reaction kinetics during the dissolution process (Supplementary Fig. 31).

## Analysis of the potential scalability of EECDs

For industrial applications, the production of large-area devices poses challenges, especially when employing intricate deposition or printing processes, which inevitably result in diminished yields and elevated production costs. In contrast, the streamlined design of EECD eliminates the need for these complex processes, offering a more efficient and practical solution. In addition, the high square resistance of the transparent electrode also presents a significant challenge in the development of large-area devices. We utilized a previously established voltage distribution equation to calculate the voltage drop ($\Delta V$) across a square electrode[18] (Eq. (3)). The effective voltage for deposition at the center of the window is 1.57 V due to Ohmic drop (Supplementary Fig. 32). Thus, efficient switching is ensured by the voltage applied to the center part, as the deposition process begins at 1.13 V, which is the potential indicated in the cyclic voltammetry (CV) curve of the EECD (Supplementary Fig. 23).

$$\Delta V = \frac{J\rho}{2t} \sqrt{\left[\left(\frac{L}{2}\right)^2 - x^2\right]\left[\left(\frac{L}{2}\right)^2 - y^2\right]} \qquad (3)$$

For an EECD as large as 100 cm$^2$ (10 cm × 10 cm) with high transmittance, uniform coloring occurs throughout the stages, making it compatible with diverse application scenarios (Fig. 5a). In addition, very little transmittance difference at both the center and edge of the windows can be observed, which confirmed the uniform tinting and bleaching of the windows (Fig. 5c). This phenomenon is quite different from the reported EC techniques, which usually show an uneven distribution of color due to the high square resistance and the unsmooth surface of TCO electrodes during coloring changing, the COMSOL Multiphysics simulation result revealed the self-regulating mechanism in EECD by modeling the electric field change during the coloring process. Particularly, with the combined effects of electromigration and diffusion, the $MnO_2$ and Cu nuclei form on both electrodes and then continuously grow. As shown in Fig. 5b (also Supplementary

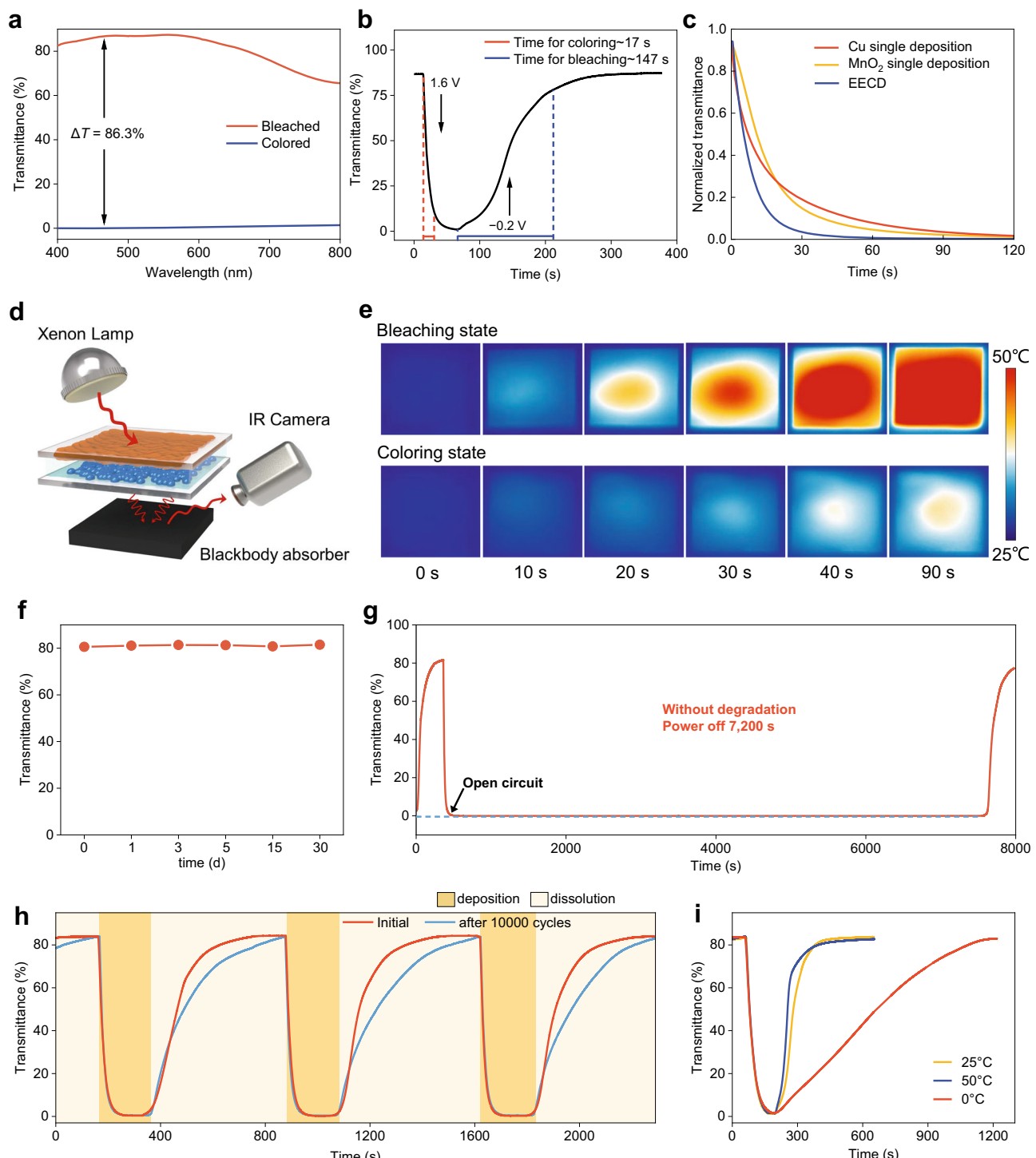

**Fig. 4 | Electrochemical and optical performance of "electrode-free" electrochromic device (EECD). a** UV-Vis transmittance spectra of EECD at the colored and bleached states across the visible spectrum of 400–800 nm. **b** A plot of transmittance over time with applied colored and bleached voltages of 1.6 V and −0.2 V, respectively. **c** Transmittance-time curves of EECD and MnO$_2$ and Cu single electrodes at 460 nm. **d** Thermal management performance measurement apparatus for EECD under xenon lamp illumination to simulate sunlight. The coloring device with deposited MnO$_2$ (brown) and Cu (blue). **e** Real-time thermal images of EECD during illumination of xenon lamp. **f** Optical transmittance of the device in its transparent state. **g** Optical transmittance variation of EECD being colored at 1.6 V for 60 s, followed by a 7200 s power-off period, and then bleached at −0.2 V, with all measurements taken at a wavelength of 460 nm. **h** Optical transmittance over time curve of EECD after 10,000 cycles, measured at the wavelength of 460 nm. **i** Optical transmittance over time curves of EECD after being aged at 50 °C for 1 h, 25 °C for 1 h, and 0 °C for 1 h, respectively. All sample measured at the wavelength of 460 nm.

Fig. 33), the modeled FTO electrode exhibited an unsmooth surface with SnO$_2$ grains of different sizes, resulting in the inhomogeneous current density and ion flux on electrodes. The electrons and ion flux preferably concentrate on the tips of SnO$_2$ grains[38]. Theoretically, the longitudinal growth of Cu into dendrites is self-accelerating, thereby short-circuiting the device[39]. However, the MnO$_2$ layer helps homogenize the ion flux and electric field on both electrodes due to its low conductivity (from $10^{-4}$ to $10^{-3}$ S m$^{-1}$), allowing the uniform plating of

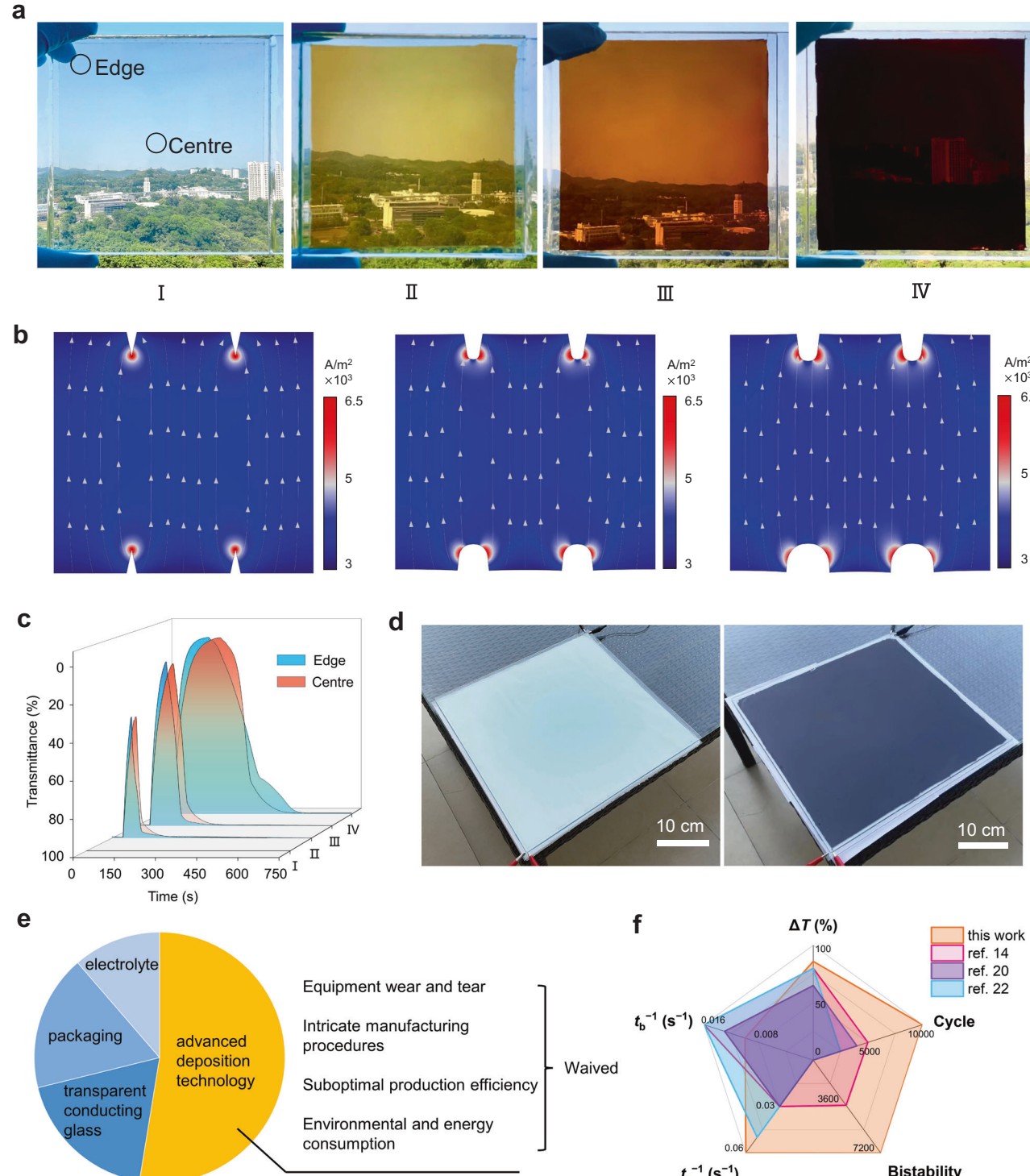

**Fig. 5 | Scalability of "electrode-free" electrochromic device (EECD).**
**a** Photographs of a well-packaged 10 cm × 10 cm EECD sample at different coloring states (I, II, III, and IV). **b** COMSOL Multiphysics simulations of the dual deposition process of $MnO_2$ and Cu from initial state to uniformly colored state (left to right). **c** Optical transmittance measured at the wavelength of 460 nm at the edge and center of the EC window (as marked in **a**) over time. **d** Photographs of a 40 × 40 cm² window in the bleached state and colored state. A piece of white paper is placed at the bottom for contrast. **e** Estimated cost composition of commercial electrochromic devices. EECD can waive a large portion of the fabrication cost. **f** A radar plot comparing the major performance indexes of optical contrast ($\Delta T$), coloration time ($t_c$), bleaching time ($t_b$), memory effect (bistability), and cycle life. Several representative works are cited in the radar plots for comparison and the results of each characteristic are schematically illustrated semi-quantitatively.

both Cu metal and $MnO_2$[40]. Benefiting from this mechanism, we can observe uniform tinting in an EECD window with a large area (40 cm × 40 cm), suggesting its promising scalability (Fig. 5d and Supplementary Fig. 34).

Last but not least, the EECD strategy can successfully bypass the complexities for manufacturing. This advantage extends beyond the economic spectrum; it manifests in the ease with which ordinary individuals can have these devices delivered and installed, without the

need for complex and expensive manufacturing facilities. In terms of cost effectiveness, the estimated cost of conventional EC windows ranges from 180 to 250 USD m$^{-2}$, in which the preparation of the EC-layer on FTO glasses, accounts for ca. 55% of the overall cost (Fig. 5e)[41,42]. In sharp contrast, EECD only consists of two pieces of FTO glass and a dilute electrolyte solution, resulting in a significantly reduced cost to ca. 80 USD m$^{-2}$ (see estimation details in Supplementary Table 2–7). To illustrate the application of EECD smart windows, a sample was integrated with a mini-solar cell panel and a compact power management module for remote controlling by a mobile phone through Wi-Fi[43] (Supplementary Fig. 35). Undoubtedly, EECD presents exceptional EC performance in various aspects, including high optical contrast, short response time, and excellent durability and bistability (Fig. 5f and Supplementary Table 8). Due to these attributes, EECDs present a robust and promising EC solution.

## Discussion

In summary, we proposed the concept of "electrode-free" EC window featuring the dual deposition/dissolution mechanism, where EC function is reflected on the corresponding cathode and anode solid-liquid transition reactions. As a proof of concept, we demonstrated a model EECD that dynamically modulates transmittance via the deposition/dissolution redox reaction of $MnO_2/Mn^{2+}$ at the cathode and $Cu^{2+}/Cu$ at the anode. The EECD delivered a high optical contrast of ca. 85% and reached 0.01% transmittance across the whole visible transmittance. The superior performance originates from the simultaneous coloring of both electrodes, corresponding to the dark neutral color of the deposited $MnO_2$ and high optical extinction coefficient contributed by the plated copper, both of which exhibit complementary light shielding spectrum. In addition, EECD exhibits outstanding cycling stability with no performance degradation after 10000 cycles and good bistability without color fading for two hours at the fully colored state and after 30 days at the bleached state. These advancements can be realized with large areas up to 1600 cm$^2$, highlighting their potential scalability. We recognize that the disparity of $t_c$ and $t_b$ may originate from the more sluggish kinetics of the dissolution reaction compared to the deposition reaction. Our analysis reveals that the dissolution of manganese dioxide encompasses three distinct stages. Thus, further work will focus on enhancement of $MnO_2$ dissolution kinetics, particularly at low temperatures, by optimizing electrolyte compositions. Furthermore, as a new concept of electrochromism, the enhancement of the universality and performance of EECDs can be achieved by exploring additional deposition/dissolution reactions and diversifying available materials.

Aiming to provide a new strategy for energy-efficient buildings and environmental sustainability, EECD embodies a pragmatic and user-friendly solution for diverse scenarios whether in crowded buildings in metropolitans or those small cottages in remote regions. With cost effectiveness, exceptional EC performance, and user-friendliness, it minimizes the budgets in building operations (including indoor lighting as well as heating and cooling loads) for existing and new buildings, to achieve all-year energy-saving with habitant comfortability. We believe that the EECD method will trigger a profound impact on energy-efficient building applications and can make significant contributions to global carbon neutrality and sustainability.

## Methods

### Materials

The following chemicals and materials are all commercially available and used as received: manganese sulfate ($MnSO_4$), copper sulfate ($CuSO_4$), sulfate acid ($H_2SO_4$), lead oxide (PbO), methanesulfonic acid ($CH_3SO_3H$, 70 wt.%), active carbon (AC), polytetrafluoroethylene (PTFE) concentrated dispersion 60 wt.% are purchased from Shanghai Maclean Biochemical Technology Co., Ltd. All other chemicals were of analytical grade and used without further purification.

### Electrochemical and EC measurements

The electrochemical tests, including cyclic voltammetry (CV) and chronoamperometry (CA), were both performed on an electrochemical workstation (VMP3, Bio-Logic) using a conventional three-electrode quartz configuration test individual electrode deposition/dissolution. The working electrode utilizing FTO glass has an immersed geometric surface area of 1.0 cm$^2$. An activated carbon (AC) and mercury-mercurous sulfate ($Hg/Hg_2SO_4$) electrode were utilized as the counter and reference electrode, respectively. The CV test of $MnO_2/Mn^{2+}$ was performed in a voltage window between 0.2 and 1.6 V against $Hg/Hg_2SO_4$ at a scan rate of 5 mV s$^{-1}$. For preparing the counter carbon electrode, AC, PTFE, and conductive black (super P) were mixed in an 8:1:1 mass ratio, and then compression film on Ti mesh and dried at 60 °C overnight. The active mass loading for AC counter electrodes was 13–15 mg cm$^{-2}$. We examined the electrochromic characteristics by a combination of the CHI660C electrochemical workstation and UV-vis–near-infrared spectrophotometer (Cary 5000, America). The optical properties were measured by using a colorimeter (CHN Spec CS-821N, Guangzhou, China).

### Material characterizations

The morphologies of the samples were researched by scanning electron microscopy (SEM, HITACHI SU8010, Japan). The crystalline information was identified by X-ray diffraction (XRD, D8 Advance, Germany) with Cu K$_\alpha$ radiation (1.54050 Å). The X-ray photoelectron spectra (XPS) were measured with Al-K$_\alpha$ radiation (50 W, 15 kV) (ESCALABSB 250 Xi). EELS maps were collected by an FEI Talos F200X equipped with an X-FEG field emission gun and a GIF Quantum 965 EELS spectrometer. HRTEM images were employed by field emission transmission electron microscopy (FEI Tecnai F30). Differential scanning calorimetry (DSC, Mettler Toledo, Swiss) was adopted to analyze the phase-transition temperature, and samples were scanned from −40 °C to 25 °C at a rate of 10 °C min$^{-1}$ under nitrogen atmosphere. Thermal images were captured using the Fluke TiX640 Infrared Camera. The simulated sunlight source was using a xenon lamp (Microsolar 300 Xenon lamp source, 320–780 nm, PerfectLight).

### Fabrication of EECD

Small-scale EECDs used for the optical performance characterization were constructed on two FTO-on-glass pieces with the size of 5 cm × 6 cm. A commercial Optically Clear Adhesive (OCA) tape stack (Dongguan Fuyin Adhesive Co. China) with the thickness of 500 μm (stacking four 125-μm-thick layers together) and width of 2 mm served as the border frame of EECD. Subsequently, the aqueous electrolyte containing 0.5 M $MnSO_4$ + 0.1 M $CuSO_4$ + 0.5 M $H_2SO_4$ was injected into the device by syringe. After injection of the electrolyte, the EECD was further sealed with the transparent Kafuter (China) K-705 rtv moisture proof insulating silicone sealant. 10 × 10 cm$^2$ windows and 40 × 40 cm$^2$ windows were fabricated in the same way.

## Data availability

The data that support the findings of this study are available in the supplementary material of this article. Additional information is available from the authors on request. Source data are provided with this paper.

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

## Acknowledgements

The authors thank the National Natural Science Foundation of China (52061160482 & 52273297), the Tsinghua University Spring Breeze Fund, Guangdong Provincial Key Laboratory of Thermal Management Engineering & Materials (2020B1212060015), Shenzhen Technical Project (GJHZ20210705143000002) and Shenzhen Geim Graphene Center for financial supports.

## Author contributions

Z.J. and Y.S., C.Y. conceived the idea and initiated the experimental design. Z.J. and L.Q. carried out the device fabrication, electrochemical experiments, and characterizations. Z.J., Y.S., and L.Q. wrote the paper. X.R., Y.Z., R.Y., L.W., D.C., and C.Y. reviewed, discussed, and edited the manuscript. C.Y. supervised the project.

## Competing interests

The authors declare no competing interests.
