## [Peer Review File · Nature Communications]

Electrochromic windows with fast response and wide dynamic range for visible-light modulation without traditional electrodesREVIEWER COMMENTS

Reviewer #1 (Remarks to the Author):

Jia et al. proposed the concept of "electrode-free" EC window (EECD) featuring the dual deposition/dissolution mechanism which showcased a novel design and excellent EC performance. The EECD device using $\text{MnO}_2/\text{Mn}^{2+}$ as cathode and Cu^{2+}/Cu as anode was able to adjust the transmittance over a wide visible spectrum range, achieving $<0.01\%$ transmittance and fast response (90% colorization in 17 seconds). In addition, EECD exhibited outstanding cycling stability with no performance degradation after 10,000 cycles and good bistability with no color fading for 2 h at the fully colored state and after 30 days at the bleached state. It was very noteworthy that EECD has demonstrated unique "self-correction" ability for color uniformity under large-area conditions ($40\text{ cm} \times 40\text{ cm}$). Then, the author also conducted detailed study on the mechanism of changes in optical properties and the mechanism of uniform discoloration over a large area caused by electrodeposition-dissolution. Last but not least, the EECD strategy significantly reduced the cost of material and manufacturing. This was a smart design and would attract the attentions of the readers of Nature Communication. However, there are some issues should be confirmed as below before the manuscript can be published.

1. Electrode was used as either of the two terminals of an electrically conducting medium and it conducted current into and out of the medium. In this paper, "Electrode-free" EC device (EECD) was described. However, FTO were used as the electrodes. Thus, "Electrode-free" should be further explained.

2. Why was the transmission spectrum at 460 nm chosen to evaluate the EC performance of $\text{MnO}_2/\text{Mn}^{2+}$? Why was the transmission spectrum at 500 nm chosen to evaluate the EC performance of Cu^{2+}/Cu ?

3. How to perform CV testing on $\text{MnO}_2/\text{Mn}^{2+}$ by using three electrode system on FTO? Please provide a detailed description in the testing method.

4. In Supplementary Fig. 10, the transmittance of the first loop after conducting electricity was lower than the original level, if so, how comes that the transmittance of the beginning of the following two loops the same as the transmittance level of the first loop at the very beginning.

5. The legend of Supplementary Fig. 20 was wrong. The figure included the full spectrum range from 400-800nm rather than just the spectrum under 460 nm.

6. In Fig. 1a, the testing conditions between III and IV were not well explained in the description of the figure. Please mark the two test conditions in the figure or in content.

7. In the content, the authors mentioned "narrow optical contrast" and "wide optical contrast". It is not accurate to use wide and narrow here. Please refer to the relevant literature to find the appropriate expression.

8. In Fig. 2c, the position corresponding to electrodeposition for 180s should be marked. In Fig. 2d, why was the transmittance so high after 600s of electrical stimulation, but the transmittance was low without stimulation?

9. In Fig. 3e, the characteristic peak of the red line HMnO_2 was not obvious, was this a normal situation?

10. Why was the oxidation-reduction peak of $\text{MnO}_2/\text{Mn}^{2+}$ in Fig. 3a so different from the oxidation-reduction peak in Supplementary Fig. 16?

11. In Supplementary Fig. 16, why did $\text{PbO}_2/\text{Pb}^{2+}$ only have a broad reduction peak and its oxidation peak was not obvious? In Supplementary Fig. 17, why did Cu^{2+}/Cu only have a broad oxidation peak and its reduction peak was not obvious?

12. Both Fig. 2e and 4b should label the time and size of the voltage applied. It can be an arrow like the one in Fig. 2a of Nat. Commun. 10, 1559 (2019).

13. The coloring efficiency mentioned in Page 7 is an important parameter to measure electrochromic performance. However, the coloring efficiency in this article is different from the coloring efficiency defined by electrochromism, which can easily lead to misunderstandings.

14. In Fig. 4h, why did the bleaching response time slow down obviously as the number of cycles increases, while the coloring response time changed slightly? Was this because of some structural changes of the material?

15. In Fig. 4i, why temperature change had a significant impact on the bleaching response time, but had a small impact on the coloring response time?

16. In Page 7, the authors mentioned that the conventional conjugated-polymer-based electrochromic windows that generally require a constant potential to sustain the coloring state.

However, not all traditional electrochromic windows require a constant potential to sustain the coloring state. In fact, many of them have good bistability. Thus, the description here was not appropriate.

Reviewer #2 (Remarks to the Author):

In this paper, the authors report an electrode-free EC device whose working mechanism is based on the dual deposition/dissolution of $\text{MnO}_2/\text{Mn}^{2+}$ for the cathode and Cu^{2+}/Cu for the anode. The complementary absorption spectra of the two electrodes allow very high optical contrast between the bleached and coloured states, with fast kinetics and high cycling stability.

Overall, I consider the reviewed paper to be very good, as it is an important step towards the development of stable electrochromic devices.

The manuscript is carefully prepared and contains interesting results accompanied by a complete characterisation, both from an electrochemical and optical point of view.

The results are clearly and accurately reported and discussed in a critical and reasoned manner and therefore, in my opinion, the work is worthy of publication.

I have only few comments:

- First of all, in Figure 2d there is no correspondence between the colours of the curves and the transmittances plotted as a function of time (e.g. as indicated in the legend, the "0 s" yellow curve should be the one with maximum transmittance).

- While the absence of electrodes is an advantage in terms of simplifying the manufacturing process and reducing overall cost, an electrode-free system leaves the FTO in direct contact with the electrolyte. Therefore, while in classical architectures the electrochromic layers (e.g. WO_3 and NiO) provide some protection to the transparent conductive oxide (TCO), in this case the FTO is in direct contact with the electrolyte. It is known that an acidic environment, such as the one described in the paper, can induce degradation of the FTO with negative effects on its electrical properties. The authors should make some comments on this.

- What about the coloration efficiency of the EECD?

Reviewer #3 (Remarks to the Author):

In this study, Jia et al. developed "Electrode-free" electrochromic windows with fast response and a wide dynamic range for visible-light modulation, which is interesting. However, several issues need to be addressed:

1 Please provide more detailed and useful information in the Methods section. For example, in the fabrication of the EECD, the typical composition of the electrolyte was not provided.

2 Please provide the corresponding input voltage for each image in Fig. 2a.

3 The uniformity of large area devices is still controversial. Is the powered voltage of the large-area device still 1.6V and -0.2 V? How does the transmittance change as a function of time with applied colored and bleached voltages of 1.6 V and -0.2 V at different regions (e.g., center and edge)? What are the switching times of the large-area device at different regions? Additionally, the data in Supplementary Table 1 corresponds to which region? Please provide more data for other regions.

4 It is recommended to clearly state the function of counter Cu electrode. Is it solely used to improve the switching speed, or does it serve other functions? How about the stability of ECD based on pure MnO_2 ? Please provide more discussion.

RESPONSE TO REVIEWERS' COMMENTS

We appreciate the detailed and constructive comments provided by the reviewers. The reviewer comments are laid out below in italic and blue. Our response is given in normal font, of which the important parts are highlighted in yellow.

Reviewer #1:

Jia et al. proposed the concept of “electrode-free” EC window (EECD) featuring the dual deposition/dissolution mechanism which showcased a novel design and excellent EC performance. The EECD device using MnO₂/Mn²⁺ as cathode and Cu²⁺/Cu as anode was able to adjust the transmittance over a wide visible spectrum range, achieving <0.01% transmittance and fast response (90% colorization in 17 seconds). In addition, EECD exhibited outstanding cycling stability with no performance degradation after 10,000 cycles and good bistability with no color fading for 2 h at the fully colored state and after 30 days at the bleached state. It was very noteworthy that EECD has demonstrated unique “self-correction” ability for color uniformity under large-area conditions (40 cm × 40 cm). Then, the author also conducted detailed study on the mechanism of changes in optical properties and the mechanism of uniform discoloration over a large area caused by electrodeposition-dissolution. Last but not least, the EECD strategy significantly reduced the cost of material and manufacturing. This was a smart design and would attract the attentions of the readers of Nature Communication. However, there are some issues should be confirmed as below before the manuscript can be published.

Response: We appreciate the reviewer’s positive comments. The questions and issues raised by the reviewer are addressed point by point below.

1. Electrode was used as either of the two terminals of an electrically conducting medium and it conducted current into and out of the medium. In this paper, “Electrode-free” EC device (EECD) was described. However, FTO were used as the electrodes. Thus, “Electrode-free” should be further explained.

Response: Thank you for your insightful comment. We have added the definition of the “electrode-free” electrochromic device (EECD) in the manuscript. The term

“electrode-free” denotes a distinctive characteristic of our electrochromic device, where no electrochromic material layer is present on the FTO glass surface in its initial state. We adopted this terminology to differentiate our electrochromic device (EECD) from conventional EC devices, which require intricate procedures to apply the electrochromic layer onto electrodes. To elucidate this distinction, we have revised the abstract where the EECD is introduced.

Page 1: “In this study, we introduce an innovative “electrode-free” electrochromic (EC) device, termed the EECD, which lacks an EC-layer on the electrodes during device assembling and in the bleached state. This device features a simplified fabrication process and delivers superior optical modulation. It achieves a high optical contrast ranging from 68-85% across the visible spectrum and boasts a rapid response time, reaching 90% coloring in just 17 seconds.”

2. Why was the transmission spectrum at 460 nm chosen to evaluate the EC performance of MnO₂/Mn²⁺? Why was the transmission spectrum at 500 nm chosen to evaluate the EC performance of Cu²⁺/Cu?

Response: We thank the reviewer for pointing out this discrepancy. We evaluated the electrochromic performance of both MnO₂/Mn²⁺ and Cu²⁺/Cu electrodes at the wavelength of 460 nm. In addition, we have corrected the typo in Supplementary Fig. 16, where the measurement wavelength was also 460 nm instead of 500 nm.

For the reason selecting the transmittance spectrum at 460 nm, we leveraged the wavelength at which the MnO₂/Mn²⁺ redox exhibits the most significant change in transmittance. This choice aligns with the recommendations from Abdulkhem Y. Elezzabi and his colleagues that the testing wavelength should be either at the peak of optical modulation or at the wavelength corresponding to maximum human visual sensitivity, in particularly around 550 nm (Elezzabi *et al.*, *Nanophotonics*, 2020, DOI: 10.1515/nanoph-2020-0474). In addition, considering the potential hazards of blue light on human health, as highlighted in the paper (Song *et al.*, *NPJ Aging*, 2022, DOI: 10.1038/s41514-022-00092-z), the selection of the 460 nm wavelength allows for an

evaluation not only of the material's performance but also its safety implications. We have added the related discussions in the manuscript.

Revision: “**Supplementary Fig. 16. a**, CV curve for $\text{PbO}_2/\text{Pb}^{2+}$ redox behaviors on FTO electrode at 10 mV s^{-1} in a three-electrode configuration using an electrolyte containing 3 M $\text{CH}_3\text{SO}_3\text{H}$ and 0.5 M PbO , and **b**, the corresponding optical transmittance at 460 nm through the spectroelectrochemical cell.”

Page 4: “It is worth noting that we selected the wavelength of 460 nm to assess the EC performance of $\text{MnO}_2/\text{Mn}^{2+}$. This selection is based on two main factors: the most pronounced change in transmittance observed at this wavelength and the safety concerns associated with the blue light spectrum.”

3.How to perform CV testing on $\text{MnO}_2/\text{Mn}^{2+}$ by using three electrode system on FTO? Please provide a detailed description in the testing method.

Response: Thanks for your suggestion. We have demonstrated the experimental details of CV in the Methods section. The following context is for your reference.

Revision: “The electrochemical tests, including cyclic voltammetry (CV) and chronoamperometry (CA), were both performed on an electrochemical workstation (VMP3, Bio-Logic) using a conventional three-electrode quartz configuration test individual electrode deposition/dissolution. The working electrode utilizing FTO glass has an immersed geometric surface area of 1.0 cm^2 . An activated carbon (AC) and mercury-mercurous sulfate ($\text{Hg}/\text{Hg}_2\text{SO}_4$) electrode were utilized as the counter and reference electrode, respectively. The CV test of $\text{MnO}_2/\text{Mn}^{2+}$ was performed in a voltage window between 0.2 and 1.6 V against $\text{Hg}/\text{Hg}_2\text{SO}_4$ at a scan rate of 5 mV s^{-1} . For preparing the counter carbon electrode, AC, PTFE, and conductive black (super P) were mixed in an 8:1:1 mass ratio, and then compression film on Ti mesh and dried at $60 \text{ }^\circ\text{C}$ overnight. The active mass loading for AC counter electrodes was $13\text{-}15 \text{ mg cm}^{-2}$.”

4.In Supplementary Fig. 10, the transmittance of the first loop after conducting electricity was lower

than the original level, if so, how comes that the transmittance of the beginning of the following two loops the same as the transmittance level of the first loop at the very beginning.

Response: Thanks for your insightful question. In Supplementary Fig. 10, we adjusted the horizontal axes of two curves (the 1000th and 2000th curves), offsetting them to avoid overlap. To avoid confusion, we replot the figure by comparing the curves in the same time scale. The following figure is a modified one for your reference. There are slight differences in the deposition-dissolution curves during the 1st cycle, which may relate to the activation process on the electrode occurring in the first cycles (Li *et al.*, *Adv. Funct. Mater.* 2021, DOI: 10.1002/adfm.202101579). This step will lead to the subsequent MnO₂ deposition and dissolution processes occur more readily at the FTO-MnO₂ interface.

Revision:

Supplementary Fig. 10. The transmittance spectra of the 1st, 1,000th, and 2,000th cycle of MnO₂/Mn²⁺ deposition/dissolution on FTO electrode.

5. The legend of Supplementary Fig. 20 was wrong. The figure included the full spectrum range from 400-800nm rather than just the spectrum under 460 nm.

Response: Thanks for your suggestion. We have revised the figure caption in Supplementary Fig. 20.

Revision: “**Supplementary Fig. 20.** UV-vis transmittance spectra of FTO electrode before depositing Cu (i) and after deposition at -0.1 V vs. SHE in three-electrode

configuration for 30 s (ii), 60 s (iii), and 180 s (iv) using electrolytes containing 0.5 M H₂SO₄, 0.5 M MnSO₄, and 0.1 M CuSO₄ respectively, in the wavelength range between 400 nm and 800 nm.”

6. In Fig. 1a, the testing conditions between III and IV were not well explained in the description of the figure. Please mark the two test conditions in the figure or in content.

Response: Thanks for your suggestion. We have updated the manuscript accordingly.

Revision: “**Fig. 1 Schematic structure and working mechanism of EECD. a,** The structures of three types of EC devices in the sequence of increased fabrication complexity, from conventional EC device containing both EC layer and counter electrode (right) to “anode-free” EC device with single counter electrode (middle), and to EECD with no electrodes (left). The photographic images of EECD were taken during the electrolyte injection process (I and II, where the red arrows point toward the liquid levels) and at different colored states after applying 1.6 V (III and IV).”

7. In the content, the authors mentioned “narrow optical contrast” and “wide optical contrast”. It is not accurate to use wide and narrow here. Please refer to the relevant literature to find the appropriate expression.

Response: Thanks a lot for your keen suggestion. We have revised the expressions in the manuscript. The following context is for your reference.

Revision:

1). “Electrochromic (EC) devices represent an emerging energy-saving technology, exhibiting the capability to dynamically modulate light and heat transmittance. Despite their promising potential, the commercialization of EC devices faces substantial impediments such as high cost, intricate fabrication process, and low optical contrast inherent in conventional EC materials relying on the ion insertion/extraction mechanism.”

2). “Furthermore, the transmittance spectra of the Cu electrode in one cycle demonstrated the Cu²⁺/Cu redox was suitable as anode chemistry in EECDs with the advantages of fast response, good reversibility, and high optical contrast

(Supplementary Fig. 21).”

8. In Fig. 2c, the position corresponding to electrodeposition for 180s should be marked. In Fig. 2d, why was the transmittance so high after 600s of electrical stimulation, but the transmittance was low without stimulation?

Response: Thanks for your kind suggestions. We have revised the figures in the manuscript accordingly. In Fig. 2c, we have marked the coloration time. In Fig. 3d, the transmittance spectra are incorrectly labeled. We have corrected the figure, where the MnO₂ electrode delivered a lower transmittance at a longer coloration time. The following figure is provided for your reference after the correction.

Revision:

Fig. 2c Peak x y color space (CIE 1931) results of MnO₂/FTO electrode at different charging times (0 s to 600 s).

Fig. 2d UV-Vis transmittance spectra of MnO₂/FTO electrode charging at a potential of 1.6 V vs. SHE for 0 s, 30 s, 180 s, 300 s, and 600 s.

9. In Fig. 3e, the characteristic peak of the red line HMnO₂ was not obvious, was this a normal situation?

Response: Thanks for your question. A similar phenomenon of the subtle peak of HMnO₂ was observed in the dissolution process of MnO₂ in a work reported by Dongliang Chao *et al.* (*Angew. Chem. Int. Ed.* 2019, DOI: 10.1002/anie.201904174). We postulate that the insertion of H⁺ ions, which have a relatively small radius, does not significantly alter the interplanar spacing within MnO₂. This could account for the less pronounced variations observed in the XRD peaks. We have added the related discussion in the manuscript. We appreciate your attention to this issue.

Page 5: “Furthermore, we conducted XRD and high-resolution transmission electron microscopy (HRTEM) to investigate the phase evolution (Fig. 3e, f). In stage II and III, a new product appeared with the XRD pattern fitted to HMnO₂ (JCPDS 18-0804) while the lattice fringes of 0.46 and 0.22 nm corresponding to the *d* spaces of the (002) and (312) planes in tetragonal HMnO₂. The unpronounced variations in XRD peaks relate to the slight distortion of the interplanar spacing of MnO₂ during H⁺ insertion.”

10. Why was the oxidation-reduction peak of MnO₂/Mn²⁺ in Fig. 3a so different from the oxidation-reduction peak in Supplementary Fig. 16?

Response: Thank you for your question. The CV profile has a close relationship with the reaction route of the materials, depending on the Gibbs free energy of the products in reactions. The difference between $\text{MnO}_2/\text{Mn}^{2+}$ and $\text{PbO}_2/\text{Pb}^{2+}$ primarily resides in the dissolution process. The dissolution of MnO_2 proceeds via two steps: (1) insertion of protons into MnO_2 to form HMnO_2 with high electrochemical stability (Yang *et al.*, *Energy Environ. Sci.* 2023, DOI: 10.1039/d3ee00018d) and (2) dissolution of HMnO_2 into Mn^{2+} , corresponding to the cathodic peak at 1 V and 0.4 V, respectively. In contrast, PbO_2 is directly reduced into Pb^{2+} in the acidic electrolyte at 0 V, and thus it makes difference in the CV curves. To make it more clear, we have supplemented discussion on the differences between the $\text{MnO}_2/\text{Mn}^{2+}$ and $\text{PbO}_2/\text{Pb}^{2+}$ redox reactions.

Page 6: “On the other hand, the $\text{PbO}_2/\text{Pb}^{2+}$ redox pair demonstrated some different properties from the $\text{MnO}_2/\text{Mn}^{2+}$ pair. For example, the deposition and dissolution of PbO_2 occurs at a lower potential range between -0.4 V and 0.4 V than the MnO_2 counterpart. More importantly, different from the $\text{MnO}_2/\text{Mn}^{2+}$ redox pair, the $\text{PbO}_2/\text{Pb}^{2+}$ redox pair only display one cathodic peak in the CV curve, referring to the direct reduction of PbO_2 into Pb^{2+} . Such characteristics have a close relationship with the fast and highly reversible tinting and bleaching processes of the $\text{PbO}_2/\text{Pb}^{2+}$ redox (Supplementary Fig. 16).”

11. In Supplementary Fig. 16, why did $\text{PbO}_2/\text{Pb}^{2+}$ only have a broad reduction peak and its oxidation peak was not obvious? In Supplementary Fig. 17, why did Cu^{2+}/Cu only have a broad oxidation peak and its reduction peak was not obvious?

Response: Thank you for your question. The features in the CV of $\text{PbO}_2/\text{Pb}^{2+}$ and Cu^{2+}/Cu redox pair originate from their deposition/dissolution mechanism. The peaks in CV appear due to the mass transfer of the reaction product reaching the limit. The deposition mechanism enables the refreshing electrode/electrolyte interface that enables efficient transport of charge carriers and reaction products. In addition, we set the potential limits where the transmittance of both electrodes in CV decreases to a minimal value, without initiating the undesired oxygen evolution reaction at the anode and hydrogen evolution reaction at the cathode. Therefore, the electric currents of PbO_2

deposition in the anodic scan and Cu deposition in the cathodic scan increase at a higher and lower potential, respectively. This is consistent with their incomplete oxidation and reduction peaks observed in the cyclic CV. In contrast, the dissolution reaction between PbO₂ and Cu is subject to mass transfer limitations, which stem from the limited mass of the reactants present on the electrodes, explaining the well-defined anodic and cathodic peaks in the CV of PbO₂/Pb²⁺ and Cu²⁺/Cu redox, respectively.

12. Both Fig. 2e and 4b should label the time and size of the voltage applied. It can be an arrow like the one in Fig. 2a of Nat. Commun. 10, 1559 (2019).

Response: Thanks for your suggestion. We had previously noted this work and had cited it in the previous version (ref No.36) We have updated the manuscript accordingly. The following figure is provided for your reference after the correction.

Revision:

Fig. 2e A plot illustrating the transmittance at 460 nm over time, with the application of deposition and dissolution potentials at 1.6 V and 0.2 V, respectively.

Fig. 4b A plot of transmittance over time with applied colored and bleached voltages of 1.6 V and -0.2 V, respectively.

13. The coloring efficiency mentioned in Page 7 is an important parameter to measure electrochromic performance. However, the coloring efficiency in this article is different from the coloring efficiency defined by electrochromism, which can easily lead to misunderstandings.

Response: Thanks for your suggestion. We have revised these expressions in the updated manuscript to make it clearer for readers. The following context is for your reference.

Original:

1). “These findings suggest that the H_2SO_4 addition facilitates the dissolution of MnO_2 . On the other hand, the coloring efficiency, namely the MnO_2 deposition rate, becomes more sluggish in acidic environments, as evidenced by the longer coloring time and lower deposition currents.”

2). “Furthermore, the EECD demonstrated a rapid and reversible optical response, with a short t_c of 17 seconds and a corresponding t_b of 147 seconds since the collective coloring of MnO_2 and Cu electrodes in EECD promotes the coloring efficiency (Fig. 4b). Fig. 4c further illustrated that EECD delivered a shorter coloring time of 60 s than that of ca. 120 s for a single MnO_2 or Cu electrode to reach the 0.01% transmittance.”

Revision:

1). “These findings suggest that the introduction of H_2SO_4 can facilitate the dissolution

of MnO₂. But **the MnO₂ deposition rate** becomes more sluggish in acidic environments, as evidenced by the longer coloring time and lower deposition currents.”

2). “Furthermore, the EECD demonstrated a rapid and reversible optical response, with a short t_c of 17 seconds and a corresponding t_b of 147 seconds since the collective deposition of MnO₂ and Cu electrodes in EECD enhances **the coloring rate** (Fig. 4b). Fig. 4c further illustrated that EECD delivered a shorter coloring time of 60 s than that of ca. 120 s for a single MnO₂ or Cu electrode to reach the 0.01% transmittance.”

14. In Fig. 4h, why did the bleaching response time slow down obviously as the number of cycles increases, while the coloring response time changed slightly? Was this because of some structural changes of the material?

Response: Thank you for your insightful question. We postulated that the longer bleaching response time at the later cycles is related to the consumption of protons throughout the cycles from the inevitable hydrogen evolution reaction in the highly acidic electrolyte. As shown in Supplementary Fig. 2, we have proven that a lower concentration of H₂SO₄ results in a longer bleaching time. On the other hand, we observed a higher pH value of the electrolyte after 5000 cycles than before cycling (0.70 vs. 0.63) (updated as Supplementary Fig. 29, as shown below). We have supplemented discussions on the fading mechanism of bleaching time in the manuscript.

Supplementary Fig. 29. pH value of the electrolyte **a**, before cycling and **b**, after 5000 cycles.

Page 8: “Cycling durability is also a key performance parameter for evaluating EC windows. After 10000 coloring-bleaching cycles, EECD exhibits a stable EC performance with no degradation of optical contrast but only a slight decrease in response time (Fig. 4h and Supplementary Fig. 28). We postulated that the longer response time relates to the slightly increased pH value of the electrolyte after 10000 cycles, which originates from the inevitable hydrogen evolution in the highly acidic electrolyte (Supplementary Fig. 29). The previous analysis has shown that the concentration of protons is very important for the dissolution kinetics. When there are fewer protons, it takes longer time for bleaching.”

15. In Fig. 4i, why temperature change had a significant impact on the bleaching response time, but had a small impact on the coloring response time?

Response: Thanks for your comment. The bleaching and coloring response time is dictated by the reaction kinetics. To unveil the detailed mechanism, we performed the electrochemical impedance spectroscopy (EIS) on the MnO₂ electrode at the colored state and bleached state and different temperatures. At 25 °C, the MnO₂ electrode at the colored state exhibits a larger charge transfer impedance (R_{ct}) than that at the bleached state, indicating the poorer kinetics of the dissolution process, consistent with the longer bleaching time than the coloring time. More importantly, the low-temperature condition aggravates the kinetics problem in the dissolution process, dramatically increasing the R_{ct} from 20 Ω to 150 Ω . In contrast, the R_{ct} of the bleached MnO₂ electrode slightly increases from 4 Ω to 11 Ω at 0 °C (updated as Supplementary Fig. 31, as shown below). Therefore, the lower temperature has a greater impact on the bleaching response time than on the coloring response time. We have supplemented related discussions to the manuscript.

Supplementary Fig. 31 Electrochemical impedance spectroscopy (EIS) profiles of MnO_2 at **a**, the bleached state, and **b**, the colored state at different temperatures.

Page 8: “Notably, the coloring performance barely changed at the various temperatures while the bleaching time increased from 145 s to 760 s when the temperature decreased from room temperature to 0 °C. This discrepancy is dictated by the different reaction kinetics of the coloring and bleaching processes. The electrochemical impedance spectroscopy (EIS) results indicate that the dissolution kinetics of MnO_2 are slower than its deposition, and the low-temperature conditions aggravate the reaction kinetics during the dissolution process (Supplementary Fig. 31).”

16. In Page 7, the authors mentioned that the conventional conjugated-polymer-based electrochromic windows that generally require a constant potential to sustain the coloring state. However, not all traditional electrochromic windows require a constant potential to sustain the coloring state. In fact, many of them have good bistability. Thus, the description here was not appropriate.

Response: Thanks for your valuable comment. We have revised the statements in the manuscript. The following context is for your reference.

Page 7: “In the idling state, the transmittance stability of EC devices in both the bleached and colored stages, namely, bistability, is crucial to their energy efficiency. The EEC samples maintained high transmittance for over 30 days in the bleached state (Fig. 4f). More importantly, the deposited MnO_2 and Cu can endure the acidic

electrolyte for 2 h without transmittance degradation in the colored state (Fig. 4g and Supplementary Fig. 25). Therefore, the operation of EECD is more energy-efficient compared to many reported EC windows that require constant potential to maintain the colored state.”

Reviewer #2

In this paper, the authors report an electrode-free EC device whose working mechanism is based on the dual deposition/dissolution of $\text{MnO}_2/\text{Mn}^{2+}$ for the cathode and Cu^{2+}/Cu for the anode. The complementary absorption spectra of the two electrodes allow very high optical contrast between the bleached and coloured states, with fast kinetics and high cycling stability.

Overall, I consider the reviewed paper to be very good, as it is an important step towards the development of stable electrochromic devices.

The manuscript is carefully prepared and contains interesting results accompanied by a complete characterisation, both from an electrochemical and optical point of view.

The results are clearly and accurately reported and discussed in a critical and reasoned manner and therefore, in my opinion, the work is worthy of publication. I have only few comments:

Response: We appreciate the reviewer's positive comments. The questions and issues raised by the reviewer are addressed point by point below.

1. First of all, in Figure 2d there is no correspondence between the colours of the curves and the transmittances plotted as a function of time (e.g. as indicated in the legend, the "0 s" yellow curve should be the one with maximum transmittance).

Response: Thank you for pointing out this error. We have corrected the typos in Figure 2d. The following figure is provided for your reference after the correction.

Fig. 2d UV-Vis transmittance spectra of MnO_2/FTO electrode charging at a potential of 1.6 V vs. SHE for 0 s, 30 s, 180 s, 300 s, and 600 s

2. While the absence of electrodes is an advantage in terms of simplifying the manufacturing process and reducing overall cost, an electrode-free system leaves the FTO in direct contact with the electrolyte. Therefore, while in classical architectures the electrochromic layers (e.g. WO₃ and NiO) provide some protection to the transparent conductive oxide (TCO), in this case the FTO is in direct contact with the electrolyte. It is known that an acidic environment, such as the one described in the paper, can induce degradation of the FTO with negative effects on its electrical properties. The authors should make some comments on this.

Response: Thank you for your insightful comment. Indeed, acidic corrosion poses a significant challenge for TCOs. This is also one of the pain points for many aqueous redox couples in the application of electrochromism. Developing deposition/dissolution electrodes compatible with organic solvent systems will be a highly beneficial direction for future research. In this manuscript, we use an aqueous redox couple as a proof of concept, hence the transparent electrode chosen should also have good corrosion resistance. Herein, fluorine-doped tin oxide (FTO) exhibits superior stability in acids, which is widely reported in the literature (McGehee *et al.*, *Joule*, 2020, DOI: 10.1016/j.joule.2020.05.008). Geiger *et al.* (*Scientific Report*, 2017, DOI: 10.1038/s41598-017-04079-9) reported that FTO is stable in the potential window $-0.34 \text{ V}_{\text{RHE}} < E < 2.7 \text{ V}_{\text{RHE}}$ in electrolytes containing H₂SO₄, which is safe for the application of both Cu²⁺/Cu (+340 mV vs. SHE) and MnO₂/Mn²⁺ (+1224 mV vs. SHE) redox reactions in our EECD. We have added the discussion to manuscript.

Page 3: “Before fabricating a full device, we assess the EC performance of the MnO₂/Mn²⁺ reaction by analyzing the ultraviolet-visible (UV-vis) transmittance spectra at 460 nm of a series of electrolytes containing 0.5 M MnSO₄ + x M H₂SO₄ (x = 0, 0.3, 0.5, 1). In this acidic condition, we select fluorine doped tin oxide (FTO) glass as transparent conductive substrate due to its excellent stability in acidic solutions.”

3. What about the coloration efficiency of the EECD?

Response: Thank you for your question. Coloration efficiency (CE) measures the optical modulation capability brought by each charge. For EECD with deposition-dissolution mechanism, the coloring efficiency must be related to its nucleation process

and heteroepitaxial growth. The EECD delivers a CE value of $15.3 \text{ cm}^2/\text{C}$ (updated as Supplementary Fig. 24, as shown below). We have added the related discussions in the manuscript.

Supplementary Fig. 24. Coloration efficiency of the as-assembled EECD.

Page 7: “Fig. 4c further illustrated that EECD delivered a shorter coloring time of 60 s than that of ca. 120 s for a single MnO_2 or Cu electrode to reach the 0.01% transmittance.

The coloration efficiency (CE) of the EECD is about $15.3 \text{ cm}^2 \text{ C}^{-1}$ at 460 nm (Supplementary Fig. 24).”

Reviewer #3:

In this study, Jia et al. developed "Electrode-free" electrochromic windows with fast response and a wide dynamic range for visible-light modulation, which is interesting. However, several issues need to be addressed:

Response: We appreciate the reviewer's positive comments. The questions and issues raised by the reviewer are addressed point by point below.

1 Please provide more detailed and useful information in the Methods section. For example, in the fabrication of the EECD, the typical composition of the electrolyte was not provided.

Response: Thanks for your kind suggestion. We have added more experiment details in the Methods section. The following context is for your reference.

Revision: “**Fabrication of EECDs.** Small-scale EECDs used for the optical performance characterization were constructed on two FTO-on-glass pieces with the size of 5 cm × 6 cm. A commercial Optically Clear Adhesive (OCA) tape stack (Dongguan Fuyin Adhesive Co. China) with the thickness of 500 μm (stacking four 125 μm-thick layers together) and width of 2 mm served as the border frame of EECD. Subsequently, the aqueous electrolyte containing 0.5 M MnSO₄ + 0.1 M CuSO₄ + 0.5 M H₂SO₄ was injected into the device by syringe. After injection of the electrolyte, the EECD was further sealed with the transparent Kafuter (China) K-705 rtv moisture proof insulating silicone sealant. 10×10 cm² windows and 40×40 cm² windows were fabricated in the same way.”

2 Please provide the corresponding input voltage for each image in Fig. 2a.

Response: Thanks for your comment. In Fig. 2a, we have applied a constant voltage of 1.6 V for the coloration process and 0.2 V for the bleaching process. For your reference, we have included the input voltages directly in the figure to enhance clarity. Please find the corrected figure provided below.

Fig. 2 Electrochromic performance of the $\text{MnO}_2/\text{Mn}^{2+}$ redox reaction. a, Photographs of MnO_2/FTO electrode, featuring color changes at different deposition/dissolution stages.

3. *The uniformity of large area devices is still controversial. Is the powered voltage of the large-area device still 1.6V and -0.2 V? How does the transmittance change as a function of time with applied colored and bleached voltages of 1.6 V and -0.2 V at different regions (e.g., center and edge)? What are the switching times of the large-area device at different regions? Additionally, the data in Supplementary Table 1 corresponds to which region? Please provide more data for other regions.*

Response: Thank you for your insightful comments. We applied the same coloration and bleaching voltage of 1.6 V and -0.2 V for the large-area device. To examine the uniformity of voltage in the $10\text{ cm} \times 10\text{ cm}$ EECD, we employed the equation proposed by Michael D. McGehee and his colleagues (*ACS Energy Letter*, 2018, DOI: 10.1021/acsenergylett.8b01781).

$$\Delta V = \frac{J\rho}{2t} \sqrt{\left[\left(\frac{L}{2}\right)^2 - x^2\right] \left[\left(\frac{L}{2}\right)^2 - y^2\right]}$$

where J is the current density, ρ is the resistivity of FTO glass ($\sim 8 \times 10^{-4} \Omega \cdot \text{cm}$), t is the film thickness of FTO coating ($\sim 1 \mu\text{m}$), L is the electrode length, and x and y are positions on the FTO surface defined by a Cartesian coordinate system with the origin at the geometric center of the FTO glass. The parameter J represents the maximum current density experimentally needed to switch EECD at 1.6 V, which was determined to be 0.24 mA cm^{-2} . The calculated ΔV is 0.03 V between the edge and the center, which is consistent with the value from literature (Barile *et al.*, *Joule*, 2024, DOI:

10.1016/j.joule.2024.01.023) and the effective voltage for deposition at the center of the window is 1.57 V due to Ohmic drop (Supplementary Fig. 33). Thus, efficient switching is ensured by the voltage applied to the center part, as the deposition process begins at 1.13 V, which is the potential indicated in the cyclic voltammetry (CV) curve of the EECD (Supplementary Fig. 23). This is corroborated by Fig. 5c in the manuscript, where over the same duration, minimal differences are observed in the bleaching and coloring processes between the center and edges of the window.

For the $40 \times 40 \text{ cm}^2$ demonstration we presented, the calculated ΔV from the edge to the center is approximately 0.26 V. While ΔV falls within the tolerance range of the EECD, and the initial deposition rate at the center region notably lags behind that at the edges. On the other hand, the deposition time for large-area devices lengthens, which in turn makes the deposition process mass-transfer controlled. Consequently, this leads to a significant reduction in the deposition current. Furthermore, as per Formula (1), this leads to a decrease in ΔV , contributing to a more uniform coloring of the device over time. Although the effect is not as ideal as it is in the 100 cm^2 case, we have still demonstrated the potential for its operation in extremely large areas. As shown in Figure R1, it takes over 2.5 hours to achieve the complete coloring with the size of $40 \times 40 \text{ cm}^2$. Notably, response time, which is less-than-ideal, is primarily attributed to the inherent limitations of the sheet resistance in the transparent conductive electrode. Furthermore, accurate monitoring of the coloring and bleaching times for super-large-area devices was not feasible due to specific constraints associated with our UV-vis-near-infrared spectrophotometer.

We recognize the limitations and challenges associated with enhancing the uniformity and response time of large-area devices. In response, we are actively seeking solutions to address these issues. Your suggestions are highly valued and have been instrumental in our efforts. The manuscript has been updated to include relevant discussions, and the content provided below is for your reference..

Figure R1. The coloring process for $40 \times 40 \text{ cm}^2$ EECD.

Page 8: “For industrial applications, the production of large-area devices poses challenges, especially when employing intricate deposition or printing processes, which inevitably results in diminished yields and elevated production costs. In contrast, the streamlined design of EECD eliminates the need for these complex processes, offering a more efficient and practical solution. In addition, the high square resistance of the transparent electrode also presents a significant challenge in the development of large-area devices. We utilized a previously established voltage distribution equation to calculate the voltage drop (ΔV) across a square electrode¹⁸ (Eq. (3)). The effective voltage for deposition at the center of the window is 1.57 V due to Ohmic drop (Supplementary Fig. 32). Thus, efficient switching is ensured by the voltage applied to the center part, as the deposition process begins at 1.13 V, which is the potential indicated in the cyclic voltammetry (CV) curve of the EECD (Supplementary Fig. 23).

$$\Delta V = \frac{J\rho}{2t} \sqrt{\left[\left(\frac{L}{2}\right)^2 - x^2\right] \left[\left(\frac{L}{2}\right)^2 - y^2\right]} \quad (3)$$

Supplementary Fig.32. Modeled voltage drop across the FTO-coated glass during EECD switching.

The voltage distribution across the FTO-coated glass is provided in Supplementary Fig. 33, which was calculated by integrating Ohm's law over the x-y plane of the FTO-coated glass, according to the following equation:

$$\Delta V = \frac{J\rho}{2t} \sqrt{\left[\left(\frac{L}{2}\right)^2 - x^2\right] \left[\left(\frac{L}{2}\right)^2 - y^2\right]} \quad (1)$$

where J is the current density, ρ is the resistivity of FTO glass ($\sim 8 \times 10^{-4} \Omega \cdot \text{cm}$), t is the film thickness of FTO coating ($\sim 1 \mu\text{m}$), L is the electrode length, and x and y are positions on the FTO surface defined by a Cartesian coordinate system with the origin at the geometric center of the FTO glass. The parameter J represents the maximum current density experimentally needed to switch EECD at 1.6 V, which was determined to be 0.24 mA cm^{-2} . The calculated ΔV is 0.03 V between the edge and the center. For the $40 \times 40 \text{ cm}^2$ demo, the ΔV from the edge to the center is about 0.26 V based on calculation.

In response to your question regarding Supplementary Table 1, we would like to clarify that the $L^*a^*b^*$ value given in Table 1 coordinates for the MnO_2/FTO electrodes, which varies with deposition time (and thus thickness), as depicted in Figure 2a. These electrodes were prepared using a three-electrode cell setup. The MnO_2/FTO electrodes

exhibit homogeneous plating in the regions that are immersed in electrolytes. The uniformity is attributed to the small size of the electrodes (1 cm × 2 cm). To improve the accuracy of the data, we measured different spots, *i.e.*, edge and center, on the electrodes and updated the Supplementary Table 1 with the average L*a*b* values of different spots on the electrodes. In addition, Fig. 2b and Fig. 2c corresponding to these data are also updated. The following updates are for your reference.

Revision:

Page 4: “At the fully bleached state, the (L*, a*, b*) coordinates are (93.74, 0.04, 2.25), and they shift to (4.81, 1.34, -0.02) at the fully colored state, indicating the transition between the colorless and black.”

Supplementary Table 1. L*, a*, and b* change with different deposition times of depositing MnO₂ on the FTO electrode.

Time (s)	L*	a*	b*
0 s	93.74	0.04	2.25
30 s	76.90	3.08	36.00
180 s	47.03	26.12	69.06
300 s	23.92	31.99	40.35
600 s	4.81	1.34	-0.02

The data are the average L*, a*, and b* values of different spots (*i.e.*, edge and center) on the MnO₂/FTO electrode.

Fig. 2b CIE L*a*b* color coordinates of the MnO₂/FTO electrode at various charging

time.

Fig. 2c Peak x, y color space (CIE 1931) results of MnO₂/FTO electrode at different charging times (0 s to 600 s).

4 It is recommended to clearly state the function of counter Cu electrode. Is it solely used to improve the switching speed, or does it serve other functions? How about the stability of ECD based on pure MnO₂? Please provide more discussion.

Response: Thanks for this good point. Solely employing the MnO₂ electrode can lead to a selective absorption of light below 550 nm (see Fig. 2d). To overcome this disadvantage, Cu electrode is selected that can effectively reduce light transmittance across all the visible light spectrum (Supplementary Fig. 22). Therefore, Cu²⁺/Cu employed as the anode for EECD can compensate for the insufficient light and thermal modulation capability of MnO₂. With this, EECD achieves a low transmittance of ca. 0% across the visible light spectrum. In addition, The simultaneous deposition of MnO₂ and Cu significantly speeds up the coloration process in EECDs. While the coloring time for individual deposition of MnO₂ and Cu is 120 seconds each, the combined deposition reduces this time to just 60 seconds (Fig. 4c).

According to your suggestion, we fabricated the EC device based on the MnO₂/Mn²⁺ redox as the EC electrode, a copper tape on a transparent glass as counter electrode, and employed the 0.5 M MnSO₄ + 0.5 M H₂SO₄ + 0.1 M CuSO₄ electrolyte. We found that the device delivered a stable EC performance for over 200 cycles,

consistent with the results from EECs (updated as Supplementary Fig. 25, as shown below). We have added the discussions to the manuscript.

Supplementary Fig. 25. Long term stability of the EC device, based on $\text{MnO}_2/\text{Mn}^{2+}$ redox for the EC electrode and a copper frame as counter electrode, was achieved by coloring at 1.55 V for 30 s and bleaching at 0 V for 90 s.

Page 7: “In addition, we fabricated the EC device based on $\text{MnO}_2/\text{Mn}^{2+}$ redox as the EC electrode, copper tape on transparent glass as the counter electrode, and the same electrolyte as a control EC device. The color change in this device is achieved solely through the deposition/dissolution of $\text{MnO}_2/\text{Mn}^{2+}$, and the Cu^{2+}/Cu reaction does not contribute to the color change. (Supplementary Fig. 25). As can be expected, this device delivered a stable EC performance.”

REVIEWERS' COMMENTS

Reviewer #1 (Remarks to the Author):

accepted without modification

Reviewer #2 (Remarks to the Author):

The authors have consistently responded to all my comments, and all the critical issues raised by the referee have been addressed and resolved.
In my opinion, the paper is now ready for publication.

Reviewer #3 (Remarks to the Author):

1. If Figure 2 represents different times at the same voltage, please label the corresponding times in the figure.
2. Please add the discussion regarding my comments to the article, especially figures (e.g. Fig. R1) , and not just in response to referees. Also, explicitly mention in the main text that the coloring of the large device requires 2.5 hours, which is crucial guidance for future research endeavors.

RESPONSE TO REVIEWERS' COMMENTS

We appreciate the detailed and constructive comments provided by the reviewers. The reviewer comments are laid out below in italic and blue. Our response is given in normal font, of which the important parts particularly for the editor's consideration are highlighted in yellow.

Reviewer #3 (Remarks to the Author):

1. If Figure 2 represents different times at the same voltage, please label the corresponding times in the figure.

Response: Thanks for your suggestion. We have now labeled the specific times directly in Figure 2 to provide clear reference points for the different measurements taken. The following figure is provided for your reference after the correction.

Revision:

Figure 1a Photographs of MnO₂/FTO electrode, featuring color changes at different deposition/dissolution stages.

2. Please add the discussion regarding my comments to the article, especially figures (e.g. Fig. R1) , and not just in response to referees. Also, explicitly mention in the main text that the coloring of the large device requires 2.5 hours, which is crucial guidance for future research endeavors.

Response: Thank you for your valuable comments and suggestions. We have revised our manuscript to incorporate your feedback.

Revision: “Benefiting from this mechanism, we can observe uniform tinting in an EEC window with a large area (40 cm × 40 cm), suggesting its promising scalability (Fig. 5d and Supplementary Fig. 34). However, complete coloring takes over 2.5 hours due to the limitations of the sheet resistance in the FTO.”

Supplementary Fig. 34. Photographs of the coloration process for 40 × 40 cm² EECD.